# The Search for Natural Inhibitors of Biofilm Formation and the Activity of the Autoinductor C6-AHL in *Klebsiella pneumoniae* ATCC 13884

**DOI:** 10.3390/biom9020049

**Published:** 2019-01-30

**Authors:** Elizabeth Cadavid, Fernando Echeverri

**Affiliations:** Grupo de Química Orgánica de Productos Naturales, Instituto de Química, Universidad de Antioquia, Calle 67 No. 53–10, Medellín 050010, Colombia; elicatoqf@gmail.com

**Keywords:** *Klebsiella pneumoniae*, resistance, biofilm, adherence, quorum sensing, inhibition, furan, pyran, phenyl-acyl compounds

## Abstract

Human nosocomial infections are common around the world. One of the main causes is the bacteria *Klebsiella pneumoniae*, which shows high rates of resistance to antibiotics. Thus, drugs with novel mechanisms of action are needed. In this work, we report the effects of various natural substances on the formation of biofilm in *Klebsiella pneumoniae*, as well as its stability. The effect of the molecules on the growth of *K. pneumoniae* was initially determined by measuring the optical density. The modification of the biofilm, the changes relating to its resistance, the effects on the bacterial adhesion to the urethral catheter and its antagonist role the hexanoyl-homoserinelactone were assessed by crystal violet, as well as by microscopy. The best effects were obtained with 3-methyl-2(5H)-furanone and 2´-hydroxycinnamic acid, which inhibited the formation of biofilm by 67.38% and 65.06%, respectively. Additionally, the remaining biofilm formed was more susceptible to gentamicin. Through microscopy examination, there were evident changes in the biofilm and adherence on the polyvinyl chloride (PVC) urethral catheter. Besides, 3-methyl-2(5H)-furanone inhibited the biofilm-forming effect of the autoinducer hexanoyl-homoserinelactone. Thus, these molecules could be developed as supplemental of antibiotics.

## 1. Introduction

Pathogenic microorganisms have developed resistance against most antibiotics currently used, including penicillins, aminoglycosides, fluoroquinolones, and cephalosporins [1]. For this reason, the World Health Organization (WHO) has launched a global alert on health risks [2]. Addressing *Klebsiella pneumoniae* is considered a critical priority; also, this bacterium, as well as *E. coli*, are often isolated from contaminated medical devices [3]. Several strategies have been proposed to overcome this problem, including the rationalization of therapies with antibiotics in humans and animals and the search for innovative molecules with new mechanisms of action, among others. The last approach has biocide effects, so these compounds frequently induce profound genetic modifications in microorganisms in the medium and the long term, thus adapting microbial resistance [4]. This resistance has already restricted the use of several antibiotics. Moreover, recently, we found in *Klebsiella pneumoniae* ATCC 13884 evidence that some medicines, such as hydrochlorothiazide and acetaminophen, promote the formation of biofilm and increase its resistance against two antibiotics, gentamicin, and ciprofloxacin [5].

The possibility of modulating bacterial communication has now been suggested as a new alternative to control the resistance of pathogens. This mechanism, called quorum sensing (QS), is carried out through the production of autoinducer molecules known as acylhomoserinelactones (AHL). In *K. pneumonia* lactones with different size side chain have been detected, such as *N*-octanoylhomoserine lactone and *N*-3-dodecanoyl-L-homoserine lactone [6], *N*-hexanoyl-homoserine lactone (C6-AHL) [7], in addition to a furanosyl borate diester [8]. The role of these autoinducer molecules is related to the induction of genes involved in the production of lytic enzymes, toxins, and exopolysaccharides, among others, for pathogenicity and virulence [9] as well as the formation of a biofilm [10]. Therefore, through the inhibition of QS, the bacterial behavior can be modified, but without biocidal effects. This inhibition can be achieved in several ways, e.g. by blocking the biosynthesis of the autoinducers, through their degradation by specific enzymes or preventing their interaction with the receptor.

In this paper, we analyzed the effects of several substances on processes associated with QS in *K. pneumoniae* ATCC 13884, specifically on the formation and stability of the biofilm, as well as its adherence to the urethral catheter, and the antagonistic role against the autoinducer C6-AHL. To achieve this, the viability of the bacterium was first tested at different concentrations of compounds, selecting those with viability higher than 85%. Then, the ability to inhibit the formation of biofilm and modify its architecture were also studied; in this way, the two most active compounds were selected. Subsequently, changes in the sensitivity of the mature biofilm to gentamicin were determined in addition to the inhibition of bacterial adhesion on a polyvinyl chloride (PVC) catheter. Finally, the role in neutralizing the effect of the natural autoinducer hexanoyl-homoserine lactone, was established too.

These molecules could use as leaders for the development of substances that reinforce the effect of antibiotics currently employed as well as to reuse antibiotics that are no longer used due to microbial resistance by the formation of biofilm.

## 2. Materials and Methods

### 2.1. Compounds

Twenty-seven compounds were selected based on their structural similarity with QS inhibitors reported in the literature (phenyl-acyl derivatives, pyridines, pyrroles, pyrazines, and pyrans, among others [11], and due to their similarity to lactone autoinducers, such as furans. These compounds (Figure 1) were purchased at Sigma (Sigma, St Louis, MO, USA); a specific code each molecule was assigned.

Furans (F): 2-methyltetrahydro-3-furanone (F1), 3-methyl-2(5H)-furanone (F2), furfural (F3), 5,6-dihydro-2(H)-pyran-2-one (F4), methyl 2-furoate (F5), 5-hydroxymethyl-2-furaldehyde (F6), 2-pentylfurane (F7), 5-ethyl-4-hydroxy-2-methyl-3(2H)-furanone (F8), 2-benzofuranyl methyl ketone (F9). Phenyl-acyl derivatives (PP): 3,5 dimethoxybenzoic acid (PP1), syringic acid (PP2), caffeic acid (PP3), 3-methoxyphenylacetic acid (PP4), 2´-hydroxycinnamic acid (95% *trans*, PP5), 4´-hydroxyphenylacetic (PP6), 3-phenyl-1-propanol (PP7), 2-methoxy-2-phenylethanol (PP8), 2-phenylethanol (PP9), methyl chavicol (Basil oil) (PP10), myristicin aldehyde (PP11), 3,4-dihydroxybenzoic acid (PP12). Pyrroles, pyridines, pyrazines (DP), others: acetylpyrazine (DP1), 5-acetyl-2-methoxypyridine (DP2), 2-acetyl-4-methylthiazole (DP3), 4-acetylpyridine (DP4), 3-acetyl-1-methylpirrole (DP5), 2-acetylpirrole (DP6). *N*-hexanoyl-L-homoserinelactone (C6-AHL) was used like an autoinductor. All assays were made by triplicate, with a stock solution of concentration 1.0 mg/mL of each compound in methanol; in all control experiments this solvent was used.

### 2.2. Bacteria

*Klebsiella pneumoniae* subsp. *rhinoscleromatis* ATCC 13884 was purchased from American Type Cell Collection (Manassas, VA, USA) cultivated in Medio Luria-Bertani (LAB M, Lancashire UK), at 37 °C. The inoculums were prepared in saline solution at 0.9%, adjusting to an optical density (OD_600nm_) of 0.05, equivalent to 10^6^ CFU/mL.

### 2.3. Equipment

In a microplate reader (Thermo Scientific, Waltham, MA, USA) the absorbance of crystal violet was determined. The spectrophotometer used was UV-Visible (Thermo Scientific, Waltham, MA, USA). Binocular microscopy was done with a Nikon Alphaphot-2 YS2, (Nikon, Tokyo, Japan), and fluorescence microscopy with a Nikon Eclipse 50*i* fluorescence microscope (Nikon, Tokyo, Japan), with a digital camera (Nikon DXM1200-F, Nikon, Tokyo, Japan). The urinary catheter was from PVC Medex (Passaic, NY, USA) caliber N 20.

### 2.4. Effects of Compounds on K. pneumoniae Viability

The viability of *K. pneumoniae* was determined analyzing the optical density (OD) at concentrations of 100, 50 and 25 µg/mL at 600 nm, using a medium/inoculum ratio of 1:1 (v/v), and incubating for 18 h at 37 °C, as described earlier [7]. The OD_600nm_ of the control without treatment to the culture containing the compounds was compared. Concentrations with a growth factor equivalent to or greater than 85% to control were selected.

### 2.5. Effects of Compounds on the Formation of Biofilm of K. pneumoniae

An inoculum of *K. pneumoniae* equivalent to 10^6^ CFU/mL was incubated at 37 °C for 30 h in Luria-Bertani (LB) agar in the microplates of 96 wells. Then, the selected substances were added at non-biocidal concentrations, as established before. After incubation was completed, the wells were washed twice with sterile water and dried at 50 °C. Later, they were dyed with 0.05% violet crystal for 10 min. The dyed biofilm was allowed to dry again, and then 250 µL f methanol were added to quantify the stain at 585 nm in a microplate reader. The absorbance of the violet crystal retained by the control sample without any treatment was taken as 100%.

### 2.6. Effects of 3-methyl-2(5H)-furanone and 2´-hydroxycinnamic Acid on the Kinetics of Biofilm Formation

An experiment was carried out using the same methodology before described to establish the optimal time of addition of the inhibitory biofilm compounds, but 3-methyl-2(5H)-furanone and 2´-hydroxycinnamic acid were added at 0, 6 and 24 h. The absorbance of crystal violet was established after 30 h of the assay.

### 2.7. Effects of Compounds on the Biofilm Architecture and the Size of the Bacterium by Scanning Electron Microscopy (SEM)

The formation of the biofilm was made in 1 cm × 1 cm glass coverslips using an inoculum of *K. pneumoniae* in saline solution at OD_600nm_ 0.05 in liquid LB medium containing the biofilm-inhibitors compounds. Thus, coverslips were incubated at 37 °C for 30 h. The fixation and drying of the material were done by heating at 37 °C for 12 h and then coated with a micronized gold microlayer. Finally, the biofilm was analyzed by Scanning Electronic Microscope, SEM (JEOL-JSM 6490LV, Peabody, MA, USA) operated at 20 kV, with 200X and 7000X of magnification.

### 2.8. Effects of 3-methyl-2(5H)-furanone on the Configuration of the Biofilm in Glass

The methodology for this test was similar to that used to evaluate the biofilm architecture by SEM. Thus, after incubation, coverslips were rinsed with water and then immersed in 0.1% acridine orange in phosphate buffer saline pH = 7.2 (PBS) for 1 min; the excess dye was rinsed off with sterile water. The stained biofilm was observed with an Episcopic-Fluorescence Attachment (EFD-3), using a B-2a filter cube coupled to the camera.

### 2.9. Effects of Compounds on the Sensitivity to Gentamicin of the Mature Biofilm

On microplate of polystyrene of 96 wells, an inoculum of *K. pneumoniae* and 3-methyl-2(5H)-furanone and 2´-hydroxycinnamic acid at non-biocide concentrations (15 µg/mL) were placed for 30 h. Then, the culture medium was renewed, and gentamicin was added to concentrations between 0.12–1.0 µg/mL. Then it was incubated again for 30 h more, and the remaining amount of the biofilm was quantified by staining with violet crystal at 0.5%. A biofilm without treatment of inhibitory molecules was studied as a control. The effectiveness of the substances towards making the biofilm more sensitive to gentamicin was calculated as:% Biofilm Sensitivity (BS) = 100 − (BS with compounds/BS control) × 100)(1)

### 2.10. Effect of Compounds on K. pneumoniae ATCC 13884 Adherence to Urethral Catheters

As reported before, the effect of the substances on the adhesion of *K. pneumoniae* to urethral catheters was evaluated [5]. Briefly, cultures of *K. pneumoniae* in the liquid LB media were incubated for 30 h at 37 °C with a piece of 5 cm, 20-caliber PVC catheter Medex^®^ in the presence of the substances at a non-biocidal concentration. Later, the catheter was washed with sterile water, then dried at 50 °C and after that dyed with violet crystal at 0.05% for 10 min. Finally, it was washed three times with water and allowed to dry at room temperature for viewing in the optical microscope at 10X magnification and analyzed by ImageJ program.

### 2.11. Effect of 3-methyl-2(5H)-furanone on the Biofilm Inductor of C6-AHL

An assay was made in microplates of 96 wells with LB medium, according to the modified method of O´Toole [12], to establish the effect of 3-methyl-2 (5H)-furanone in the capacity of C6-AHL to induce the formation of biofilm. Thus, C6-AHL at a final concentration of 0.4 µg/mL was placed in several wells. Similarly, in another wells 3-methyl-2(5H)-furanone at 15 µg/mL was added, and finally, a mixture of both compounds at the described concentrations was also deposited. Subsequently, an inoculum of K. pneumoniae adjusted to a OD_600nm_ of 0.05 UA was added to the wells, and next it was incubated at 37 °C for 30 h. The measurement of the adhered bacterial mass was made by staining with violet crystal at 0.05% for 10 min, washed with distilled water and then dried for 2 h at 50 °C, and extracted with 250 µL of methanol dry again. In a microplate reader the crystal violet absorbance (biofilm) was quantified at 585 nm.

### 2.12. Statistical Analysis

Stratigraphic Centurion software (16.2.04) (The Plains, VA, USA) was used to establish a comparison in means of the assays with a confidence level of 95%.

## 3. Results

### 3.1. Effects of Compounds on the K. pneumoniae Viability

Nearly all compounds showed viability of >85% at all assayed concentrations, including the lowest 25 µg/mL concentration, except 4´-hydroxyphenylacetic (PP6) and 3,4-dihydroxybenzoic acid (PP12) (Figure 2). Therefore, a lower level for the other experiments was selected (15 µg/mL) to avoid possible biocidal effects that could modify the results.

### 3.2. Effects of the Substances on the Formation of Biofilm of K. pneumoniae

A value of 20% was established as the minimum inhibition to determine the effectiveness of the compounds in the inhibition of biofilm formation since the control without treatment reached that value. The concentration used for each substance was previously established at 15 µg/mL. Most of the compounds did not affect the formation of biofilm since they had effects beneath the control limit selected (20%) (Figure 3).

Only two compounds were highlighted: 3-methyl-2(5H)-furanone, as it induces a biofilm inhibition of 67.38%, and the phenyl-acyl derivative 2´-hydroxycinnamic acid, with an inhibition level of 65.06%. However, the compound 2-methoxy-2-phenylethanol displayed an inducing effect of biofilm formation of 10.27%, with 2-benzofuranyl methyl ketone and 2-methyltetrahydro-3-furanone having lesser effects. In the group of pyrroles and pyridines, as well as pyrones, the results were negative.

### 3.3. Effects of 3-Methyl-2(5H)-Furanone and 2´-hydroxycinnamic acid on the Kinetics of Biofilm Formation

For this test, 3-methyl-2(5H)-furanone was selected. Due to its structure, it has a slight resemblance to the autoinducer lactones involved in QS; besides, it caused the most significant decrease in biofilm development. On the other hand, the formation of the biofilm occurs in several phases: 0 h corresponds to the adhesion phase, 6 h is the aggregation phase, and 24 h is the maturation phase. However, in this case, the formation of biofilm was significantly modified when the compound was added at the beginning of the experiment, at 0 h (Figure 4). No significant changes were observed at 6 and 24 h. Therefore, the compound 3-methyl-2(5H)-furanone can change the adhesion phase of *K. pneumoniae*, which generates a decrease in the formation of mature biofilm.

### 3.4. Effect of 3-methyl-2(5H)-furanone on the Architecture of the Biofilm of K. pneumoniae by Scanning Electron Microscopy (SEM)

In the images of the formed biofilm (Figure 5), fewer colonization areas were observed compared to the control, due to increases in the spaces between the microcolonies adhered to the material. The structure of the biofilm in the treated cultures is less compact and possibly more exposed to exogenous substances.

Regarding bacterial morphology in the biofilm, images of the colonized catheter at 7000X magnification were taken, then the sizes of 170 treated and untreated bacteria were measured. For both compounds, approximately 20% of the population measured between 800–1199 nm, 40% between 1200–1599 nm, 30% between 1600–1999 nm and in the remaining 10% the size was higher than 2000 nm. In the control without treatment, the measurements were 15.79%, between 800–1199 nm, 53.80% between 1200–1599 nm, and 28.65% between 1600–1999 nm. Only 1.75% of the bacteria were sized higher than 2000 nm. Thus, there was a close to 8% reduction in the number of larger bacteria.

### 3.5. Effects of Compounds on the Sensitivity of the Mature Biofilm of K. pneumoniae to Gentamicin

As mentioned before, 2´-hydroxycinnamic acid and 3-methyl-2(5H)-furanone are inhibitors of the formation of biofilm, with percentages higher than 60%. So, it is essential to determine the stability of the mature biofilm concerning the effect of gentamicin. Thus, when gentamicin (1.0 µg /mL) was added to the mature biofilm formed in the presence of the inhibitors the amount of biofilm remaining was reduced of 42.51% and 33.82%, respectively (Figure 6). This reduction is complementary to the inhibition that was previously determined, which was 67.38% and 65.06%, respectively.

At lower concentrations of the antibiotic, the effect remains, since at 0.12 µg/mL of gentamicin the reduction was 29% for both inhibitors. All this suggests that compounds 2´-hydroxycinnamic acid and 3-methyl-2(5H)-furanone not only inhibit the formation of biofilm but also make it more susceptible to gentamicin.

### 3.6. The Effect of 2´-hydroxycinnamic acid and 3-methyl-2(5H)-furanone on the Adherence of K. pneumoniae to the Catheter

The assays were carried out in a PVC urethral catheter by incubation for 30 h with these compounds at 15 µg/mL; as a control, a catheter without any compound was used. Three different zones in an area of approximately 17.000 µm^2^ with the ImageJ software, were analyzed (Figure 7). In the control experiment, the colonization area was 4709 pixels^2^, while in the material treated with 2´-hydroxycinnamic acid it was 1887 pixeles^2^, equivalent to a reduction of 60.15%. For the case of 3-methyl-2(5H)-furanone, the colonized area was 1525 pixeles^2^, with a decrease of 67.82%.

### 3.7. Effects of 3-Methyl-2(5H)-Furanone on the Configuration of the Biofilm in Glass

In coverslips, the biofilms formed by *K. pneumoniae* without any treatment are shown as dense groups (Figure 8). At the same time, bacterial agglomerations form well-defined three-dimensional structures. In the case of biofilm under 3-methyl-2(5H)-furanone treatment at 15 µg/mL (viability 95.90%), the bacterial density decreased notably. Also, bacterial clusters, although spread throughout the surface, form thin layers, which is evidenced by the reduction in the light intensity of the staining.

### 3.8. Effects of 3-Methyl-2(5H)-Furanone against the C6-AHL Autoinducer

Previously, we demonstrated that *K. pneumoniae* produces the autoinducer C6-AHL [7]; this compound is involved in the formation of biofilm. Therefore, it was assessed whether 3-methyl-2(5H)-furanone at 15 µg/mL could neutralize the effect of C6-AHL at the minimum inducing concentration of biofilm (0.4 µg/mL) [5]. For this, from the beginning of the experiment, a culture of *K. pneumoniae* was exposed to this substance. Figure 9 shows the usual effect of C6-AHL inducing biofilm by 15.65%, but 3-methyl-2-(5H)-furanone withdraws this effect as the biofilm was reduced by 24.65%. This effect was irreversible, because after adding a mixture of 3-methyl-2-(5H)-furanone and C6-AHL, the ability to form biofilm was not recovered.

## 4. Discussion

Quorum sensing (QS) is a mechanism of bacterial communication used to modulate bacterial virulence through the synthesis of lytic enzymes, toxins, and biofilm [13]. At the hospital level, some resistant bacteria produce biofilm within their first 24 h on medical devices, causing recurrent infections which are difficult to eradicate [2]. The formation of biofilm is one of several strategies found in bacteria to overcome biocidal effects of antibiotics because they become impermeable [14], and this is the first defense mechanism of *K. pneumoniae*, a bacterium for which the WHO has made an urgent call, seeking new and better tools for its control [2]. The blockade of QS (also called Quorum Quenching or QQ) has been proposed as an alternative to overcome bacterial resistance and to achieve effective treatments with current antibiotics [9,15]. These substances have been called pathoblockers [16], antipathogenic compounds, or antivirulence compounds and the antibacterial effect could be accomplished by inhibiting the production of autoinducer molecules or by blocking their interaction with a receptor. In this way, its virulence and the resistance mediated by the biofilm would practically be nullified. Although several natural products have been previously analyzed as inhibitors of bacterial biofilm [17], examples of their effects on *K. pneumoniae*, as well as on the change in the sensitivity to antibiotics and their interaction with autoinducers, are also scarce [18,19,20,21].

In the search for new molecules to overcome the resistance of *Klebsiella pneumoniae* initially, we evaluated the viability of 27 compounds belonging to different chemical groups such as furans and phenyl-acyl derivatives, as well as thiazoles, pyrazines and pyridines (Figure 1), among others. Almost all compounds displayed high viability, even at a concentration as low as 25 µg/mL demonstrating a non-biocide effect.

Of the eight compounds evaluated in the series of furans, only 3-methyl-2(5H)-furanone showed significant inhibition of 67.38% in the formation of biofilm. The others evaluated were practically inactive. As for the series of the phenyl-acyl derivatives, only the 2´-hydroxycinnamic acid caused a similar inhibition, 65.06%. From the derivatives of pyrrole, thiazole, and pyridine, only 2-acetylpirrole could inhibit a maximum value of 23%, an amount practically equivalent to that of the control (20%).

In conclusion, only 2´-hydroxycinnamic acid and 3-methyl-2(5H)-furanone showed a net effect on the reduction of biofilm, but also make the mature biofilm less stable to gentamicin, since there is another further reduction of biofilm to 42.51%, and 33.82% respectively. These compounds also showed a marginal effect on the formation of biofilm in the urethral catheter.

Finally, the compound 3-methyl-2 (5H)-furanone was able to neutralize the inductive effect of biofilm of C6-AHL, which irreversibly reduced by 24.65%.

There was no correlation between structure and activity, as substances as diverse as 2´-hydroxycinnamic acid and 3-methyl-2 (5H)-furanone have the same effect on biofilm formation. All this seems to indicate that both compounds possibly have a different target in the QS mechanism in *K. pneumoniae*. However, the presence of the lactone was essential for this activity, since 2-methyl-tetrahydro-3-furanone, which is an isomer of 3-methyl-2(5H)-furanone, was practically inactive as a biofilm inhibitor.

It is important to emphasize that compounds 2´-hydroxycinnamic acid and 3-methyl-2(5H)-furanone are found in foods. The first is in Cinnamomum and grapes, and the second is a whiskey scent and wine, among others. It has been proposed that the inhibitory molecules of quorum sensing present in food could contribute to avoidance of the formation of biofilm, thus reducing the risk of recurrent hospital infections and at the same time increasing the sensitivity to antibiotics [22,23]. Thus, it has been reported that garlic extract (rich in ajoene content, with anti-*QS* activity), supplied with the antibiotic tobramycin in vitro and in vivo, significantly reduced the formation of biofilm in *Pseudomonas aeruginosa* [24,25].

## 5. Conclusions

In summary, in this paper has been shown that two natural molecules present in food, 2´-hydroxycinnamic acid and 3-methyl-2(5H)-furanone, modified in vitro several processes related to QS in *K. pneumoniae*, which are involved in its resistance to antibiotics. Thus, those compounds affected the formation and the stability of the biofilm, making it more unstable but also sensitive to gentamicin. Besides, the adherence to the urethral catheter was also blocked. These molecules could be of great importance in the development of new antibiotics to fight pathogenic bacteria in a non-biocidal way. They could also used in combination therapy with the current antibiotics or incorporated into medical devices to prevent the formation of biofilm and therefore microbial resistance and spread of infections.

In case of 3-methyl-2(5H)-furanone the mode of action can be explained by interference with the autoinducer C6-AHL.

## Figures and Tables

**Figure 1 biomolecules-09-00049-f001:**
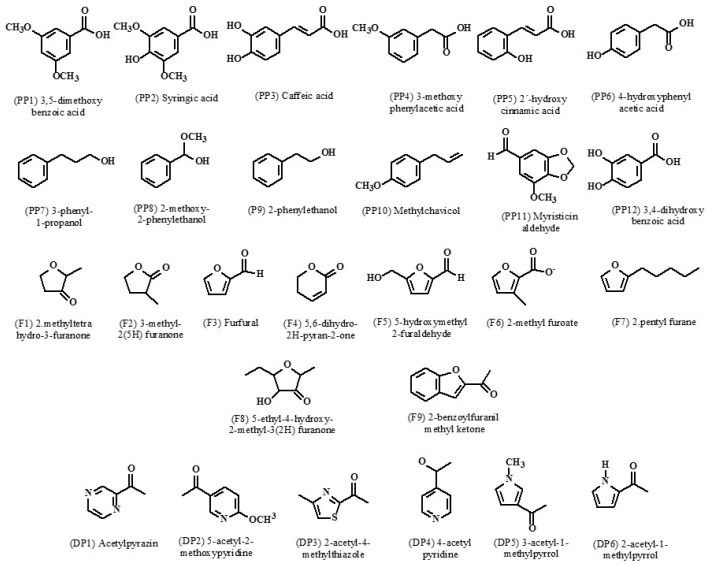
The chemical structures of compounds assayed for biofilm inhibitors.

**Figure 2 biomolecules-09-00049-f002:**
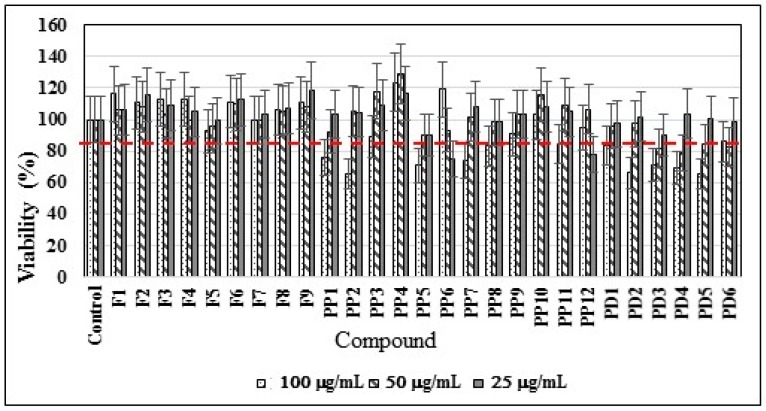
The effects of the compounds evaluated on the viability of *K. pneumoniae*. Minimum viability 85% growth was established respect to the found in the control medium without substances. See structure code in Figure 1.

**Figure 3 biomolecules-09-00049-f003:**
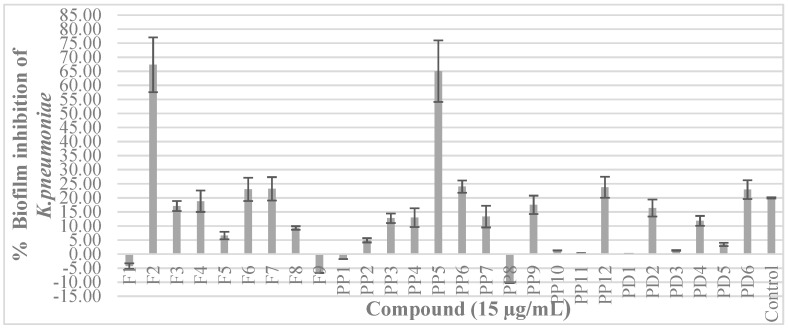
Effects of the compounds on the formation of the *K. pneumoniae* biofilm. The biofilm was obtained from an inoculum of *K. pneumoniae* in microplates of polystyrene and quantified by absorbance with violet crystal. In LB medium all substances were evaluated at 15 µg/mL (non-biocidal concentration). Negative values indicate a biofilm-inducing effect. See structure code in Figure 1.

**Figure 4 biomolecules-09-00049-f004:**
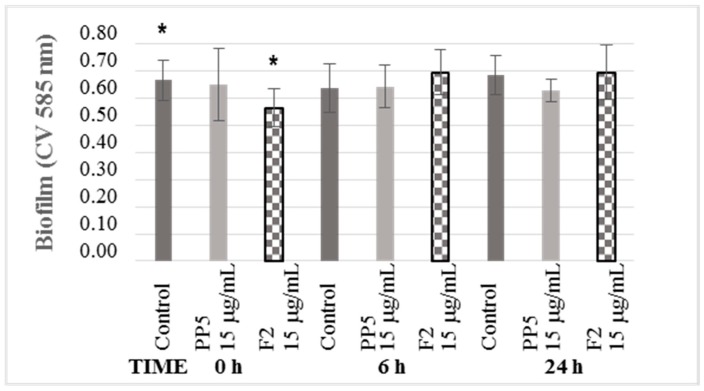
The kinetics of biofilm inhibition of the compounds. At 0, 6 and 24 h of incubation the compound was added, and the amount of biofilm formed was quantified by staining with violet crystal at 30 h. The asterisk (*) represents values lower than the average of the control of each compound concentration with statistically significant difference (* p < 0.05) (n = 5).

**Figure 5 biomolecules-09-00049-f005:**
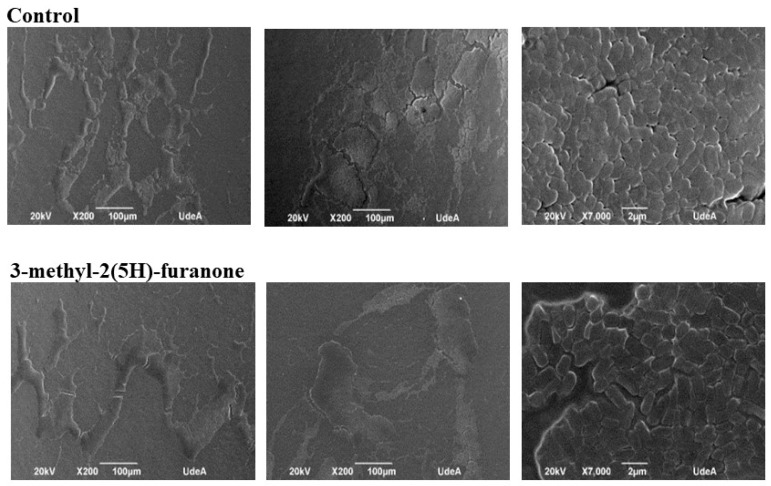
The appearance of the biofilm of *K. pneumoniae* observed with Scanning Electron Microscope (SEM) at 200X and 7000X. Upper: Culture without treatment. Bottom: Biofilm formed under the effects of 3-methyl-2(5H)-furanone. The compound reduces the colonization area and increases the interbacterial spaces.

**Figure 6 biomolecules-09-00049-f006:**
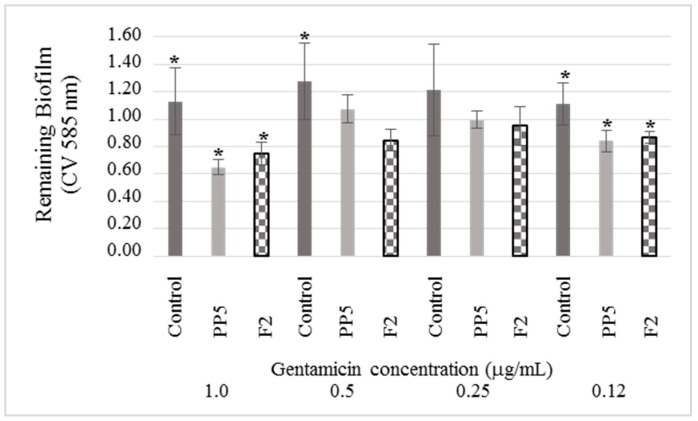
The effect of gentamicin on the elimination of mature biofilm formed under the effect of 2´-hydroxycinnamic acid and 3-methyl-2(5H)-furanone inhibitors (30 h incubation). These compounds increased the sensitivity of the biofilm of *K. pneumoniae* to gentamicin. For both substances the relative reduction of biofilm concerning the control and to each concentration of gentamicin it is appreciated. Results along with the standard deviation are presented as average values. The asterisk (*) represents values lower than the average of the control of each antibiotic concentration with statistically significant difference (* p < 0.05) (n = 5).

**Figure 7 biomolecules-09-00049-f007:**
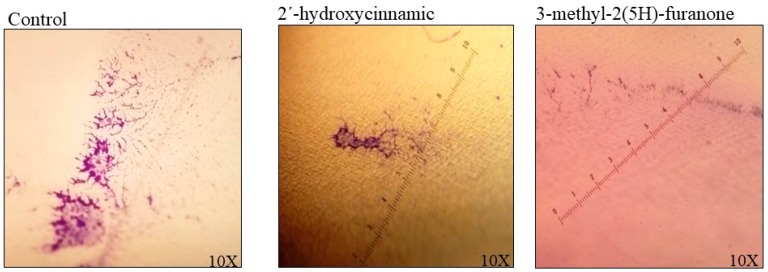
The appearance of the biofilm of *K. pneumoniae* in a polyvinyl chloride (PVC) urethral catheter. Biofilm formed at 30 h of incubation at 37 °C was stained with violet crystal at 0.05%. The size of microcolonies was determined in three fields to quantify the colonization level; each image is a catheter field with greater biofilm formation. A 60.15% reduction in colonization was observed in the presence of 2´-hydroxycinnamic acid, and a 67.62% reduction with 3-methyl-2(5H)-furanone at 15 µg/mL was observed.

**Figure 8 biomolecules-09-00049-f008:**
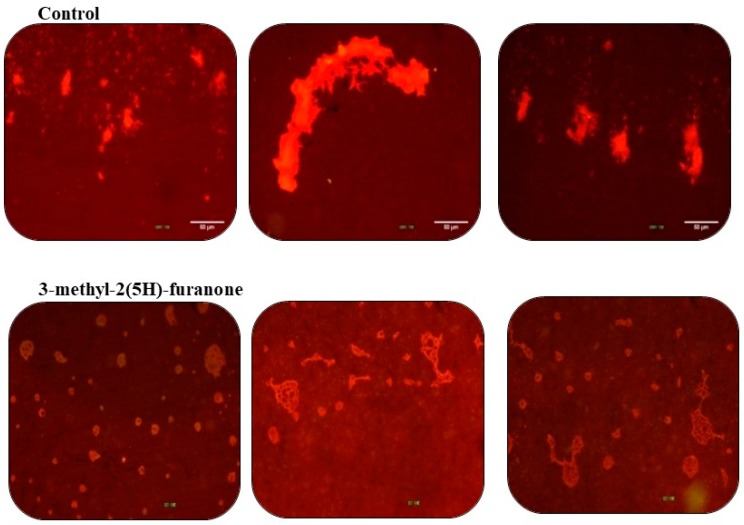
The biofilm of *K. pneumoniae* on coverslips. Upper: Biofilm formed for 30 h without treatment. Bottom: Biofilm formed under the effect of 3-methyl-2(5H)-furanone at 15 µg/mL. All biofilms were stained with acridine orange at 0.1% and observed with a fluorescence microscope with a blue filter.

**Figure 9 biomolecules-09-00049-f009:**
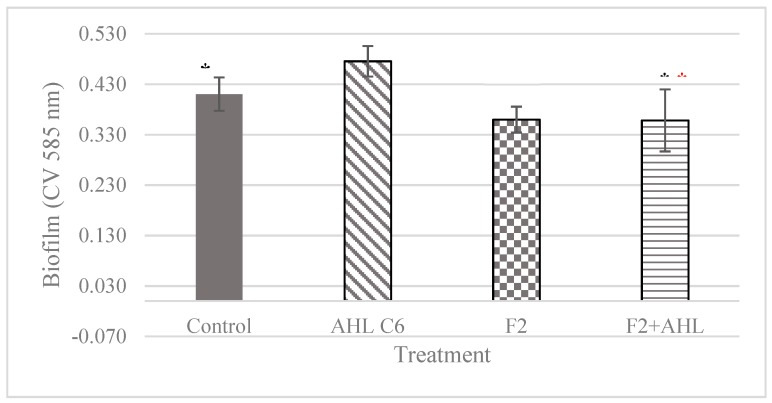
The neutralizing effect of C6-AHL autoinducer with 3-methyl-2(5H)-furanone (G2). The biofilm was formed for 30 h at 37 °C with the addition of the compounds from the beginning of the assay. The biofilm was quantified by staining with violet crystal. The black asterisks indicate statistically significant difference to the control, whereas the red asterisks indicate statistically significant difference as compared to C6-AHL.

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
