# Peer review of "The Search for Natural Inhibitors of Biofilm Formation and the Activity of the Autoinductor C6-AHL in Klebsiella pneumoniae ATCC 13884"

_biomolecules, 2019, doi:10.3390/biom9020049_

Round 1

Reviewer 1 Report

Overall, a nice study. The choices of compounds to test and application to Klebsiella are both strong points. I think it might be useful to verify a lack of impact on viability by assessing it on biofilm using a metabolic stain like MTT or XTT instead of only staining with CV. I have attached some edits I made on the manuscript--suggestions of how you might modify some of the sections. However, I do think the current version will require major editing of language and style to be clear in its information. For instance, it is not clear throughout your results section that you are using methanol as your vehicle only/control/untreated control in each of the assays, so restating it in figures or figure legends or perhaps in the text would be clearer and remind the reader.

Author Response

Due to recommendation of a referee the title of the article was changed from “Natural Inhibitors of Biofilm Formation and the Activity of the Autoinductor C6-AHL in Klebsiella pneumoniae ATCC 13884” to “The Search for Natural Inhibitors of Biofilm Formation and the Activity of the Autoinductor C6-AHL in Klebsiella pneumoniae ATCC 13884”. Similarly, there were modifications in the Introduction, Methodology, and Discussion. Finally, the original article was edited by a service of MDPI, however after that the authors added text and changed some content.

REFEREE 1

1. I think it might be useful to verify a lack of impact on viability by assessing it on biofilm using a metabolic stain like MTT or XTT instead of only staining with CV.

· The approach with CV has become a well-accepted model for biofilm formation:

o   Franklin et al. New Technologies for Studying Biofilms. Microbiol Spectr. 2015; 3(4).  doi:10.1128/microbiolspec.MB-0016-2014.

o   Lebeaux. et al. From in vitro to in vivo Models of Bacterial Biofilm-Related Infections-Pathogens. 2013, 2: 288–356. doi: 10.3390/pathogens2020288

o   Goncalvez et al. Anti-Biofilm Activity: A Function of Klebsiella pneumoniae Capsular Polysaccharide. PLoS One. 2014; 9(6): doi: 10.1371/journal.pone.0099995

2. I have attached some edits I made on the manuscript--suggestions of how you might modify some of the sections. However, I do think the current version will require major editing of language and style to be clear in its information.

Ok, those reviews have already been made

3. For instance, it is not clear throughout your results section that you are using methanol as your vehicle only/control/untreated control in each of the assays, so restating it in figures or figure legends or perhaps in the text would be clearer and remind the reader.

· The following line was added: All assays were made by triplicate, with a stock solution of concentration 1.0 mg/ml of each compound in methanol; this solvent was used in all control experiments

Reviewer 2 Report

The aim of the study is evident, but the same cannot be sayed for its execution and its presentation. You discuss positive results in the section 3.2, which are depidted in Figure 3, which is not present in the manuscript, so it is impossible to verify the statesment that you are making. I have also some doubts about the sensitivity of the mature biofilm to gentamicin. From Figure 6, it seems that at higher gentamicin concentration the remaining biofilm is more that the lower gentamicin concentrations: how can you explain this? 

Some minor remarks:

- line 95-96: LB agar in 96-well multiwell plates? Why a solid medium for a broth assay?

- line 118: probably PBS stands for Phospate Buffer Saline and not for plasma bovine serum (which I do not think is siutable as a solvent for dyes used in cell staining);

- line 122-123: please rephrase, is not clear

- line 145: I am not sure if the word "dyeing" can be used for this procedure, probably the one you were looking for was "staining" 

- line 146: why are you talking about violacein when reporting the result of a crystal violet staining?

- Figure 4: please standardize the colors in the chart, each sample has to be always of the same color;

-line 208: instead of  "formed by these inhibitors" the form which could explain better the concept could be "biofilm formed in presence of the inibitors".

Author Response

Due to recommendation of a referee the title of the article was changed from “Natural Inhibitors of Biofilm Formation and the Activity of the Autoinductor C6-AHL in Klebsiella pneumoniae ATCC 13884” to “The Search for Natural Inhibitors of Biofilm Formation and the Activity of the Autoinductor C6-AHL in Klebsiella pneumoniae ATCC 13884”. Similarly, there were modifications in the Introduction, Methodology, and Discussion. Finally, the original article was edited by a service of MDPI, however after that the authors added text and changed some content.

REFEREE 2

The aim of the study is evident, but the same cannot be sayed for its execution and its presentation.

1. You discuss positive results in the section 3.2, which are depidted in Figure 3, which is not present in the manuscript, so it is impossible to verify the statesment that you are making.

·    Figure was embedded in the text

2. I have also some doubts about the sensitivity of the mature biofilm to gentamicin. From Figure 6, it seems that at higher gentamicin concentration the remaining biofilm is more that the lower gentamicin concentrations: how can you explain this? 

• That is true in absolute terms, but we were interested in determining the change induced by both substances (PP5 and F2) in the amount of biofilm in relation to each control and each concentration of gentamicin.

Some minor remarks:

3. line 95-96: LB agar in 96-well multiwell plates? Why a solid medium for a broth assay?

·   No, liquid LB medium was used.

4. line 118: probably PBS stands for Phospate Buffer Saline and not for plasma bovine serum (which I do not think is siutable as a solvent for dyes used in cell staining);

·   Effectively, PBS is Phosphate Buffer Saline pH 7.2

5. line 122-123: please rephrase, is not clear

·   It was rewritten: On microplate of polystyrene of 96 wells an inoculum of K. pneumoniae, 3-methyl-2(5H)-furanone and 2´-hydroxycinnamic acid at non-biocide concentrations (15 µg/ml) were placed for 30 hours. Then, the culture medium was renewed, and gentamicin was added to concentrations between 0.12-1.0 µg/mL.

6. line 145: I am not sure if the word "dyeing" can be used for this procedure, probably the one you were looking for was "staining" 

·   Thanks, it was corrected

7. line 146: why are you talking about violacein when reporting the result of a crystal violet staining?

·   Ok, it was corrected

8. Figure 4: please standardize the colors in the chart, each sample has to be always of the same color;

·    Ok, it was made

9. line 208: instead of "formed by these inhibitors" the form which could explain better the concept could be "biofilm formed in presence of the inibitors".

·   Ok, it was corrected

Reviewer 3 Report

Human nosocomial infections and an increasing resistance of relevant bacteria to all sorts antibiotics constitute a serious problem globally. One of the available options, beside the invention of new classes, is to enhance and support the basic actions of existing antibiotics by supplemental actions of other compounds.  Such compounds - be it natural or synthetic - could possibly moderate the bacterial interactions with hosts, modulate inter-bacterial 'communication' and act synergistically with the known antibiotics. The Authors aimed at modifications to the biofilm formation process by a devised Klebsiella pneumoniae model and in one of its feasible applications to medical devices (an urethral catheter).

The Authors had selected 27 compounds,  structurally related to the quorum sensing inhibitors,  on the basis of the relevant literature.  The title implies a more detailed analysis of the inhibitors selected for the study.  Effectively, as indicated in the Results [lines 163-165 and 172] only two compounds were the inhibitors thoroughly tested.

My comments and some specific points:

The C6-AHL autoinductor and its role  should be briefly presented in the Introduction.

This approach was partly uncovered in the Materials and Methods section. The points 2.6, 2.8 and 2.11 were already limited to the specific compound ahead of the Results.

[lines 23 and also line 323] The use of the word “adjuvant” is not the most appropriate one in this context. These substances are “supplemental” to and enhance the action of known antibiotics.

OD (i.e. optical density), when reported, should always include the wavelength. Check also line 143.

[line 95] UFC should be replaced by CFU. Also [line 96] concentrations were “non-biocidal” (check throughout the text).

[lines 95/96] Was it LB agar or liquid medium?

[line 118] What is a “ plasma bovine serum (PBS)”?

The viability of K. pneumoniae was assessed at 3 arbitrary concentrations, with 25 μg/ml as the lowest. However another (15 μg/ml, not tested?) concentration was selected for the actual experiments [line 154]. This inconsistency should be  corrected.

Figure 3. is missing in the manuscript. [line 167]

What is the justification for the applied viability threshold (>85%). The sentence [line 153] should be re-written as it is about the “effect of compounds on the K. pneumoniae viability”.

p { margin-bottom: 0.1in; line-height: 120%; }

Author Response

Due to recommendation of a referee the title of the article was changed from “Natural Inhibitors of Biofilm Formation and the Activity of the Autoinductor C6-AHL in Klebsiella pneumoniae ATCC 13884” to “The Search for Natural Inhibitors of Biofilm Formation and the Activity of the Autoinductor C6-AHL in Klebsiella pneumoniae ATCC 13884”. Similarly, there were modifications in the Introduction, Methodology, and Discussion. Finally, the original article was edited by a service of MDPI, however after that the authors added text and changed some content.

REFEREE 3

1. The title implies a more detailed analysis of the inhibitors selected for the study.  Effectively, as indicated in the Results [lines 163-165 and 172] only two compounds were the inhibitors thoroughly tested.

·         Ok, title was changed to The Search for Natural Inhibitors of Biofilm Formation and the Activity of the Autoinductor C6-AHL in Klebsiella pneumoniae ATCC 13884

2. The C6-AHL autoinductor and its role  should be briefly presented in the Introduction. This approach was partly uncovered in the Materials and Methods section.

·         Information concerning several autoinductor lactones in Klebsiella pneumoniae was included

3. The points 2.6, 2.8 and 2.11 were already limited to the specific compound ahead of the Results.

·         More advanced analysis was carried out with furanone, due to a slight resemblance to the autoinductor lactones involved in QS; besides,these molecule caused the greatest decrease in biofilm formation.

4. [lines 23 and also line 323] The use of the word “adjuvant” is not the most appropriate one in this context. These substances are “supplemental” to and enhance the action of known antibiotics.

·         Ok, has been changed

5. OD (i.e. optical density), when reported, should always include the wavelength. Check also line 143.

·         OK, it was added

6. [line 95] UFC should be replaced by CFU.

·         It was corrected

 7. Also [line 96] concentrations were “non-biocidal” (check throughout the text).

·         It was corrected

8. [lines 95/96] Was it LB agar or liquid medium?

·         The solid medium (LB agar) was used for maintenance of the strains only; for compounds assays, liquid medium was used.

9. [line 118] What is a “ plasma bovine serum (PBS)”?

·         It was corrected to phosphate buffer saline pH 7.2

10. The viability of K. pneumoniae was assessed at 3 arbitrary concentrations, with 25 μg/ml as the lowest. However another (15 μg/ml, not tested?) concentration was selected for the actual experiments [line 154]. This inconsistency should be corrected.

·         To guarantee non-biocidal effects, the concentration of 15 μg/ml was selected, lower than 25 μg/ml previously established. Therefore, all assays were made with 15 μg/ml

11. Figure 3. is missing in the manuscript. [line 167]

·         It was corrected, since it was embedded in the text

12. What is the justification for the applied viability threshold (>85%).

·         This was an arbitrary value that was selected because we wanted to ensure a high viability to eliminate effects biocides effects

13. The sentence [line 153] should be re-written as it is about the “effect of compounds on the K. pneumoniae viability”.

·         Ok, it was corrected

Reviewer 4 Report

This study by Cadavid and Echeverri aims to characterize the effects of naturally-derived compounds on biofilm formation and antibiotic resistance in Klebsiella pneumoniae. While the work has the potential to be significant, the study when considered in its present form is missing experiments I think would be necessary to better support the conclusions.

1) While biomass (as measured by CV) can provide support for effects on biofilm formation, to truly quantify decreases in biofilm formation, it is important to measure the CFUs. Biomass and viability are often correlated, but not always. If the authors would rather not do the CFU calculations, I would recommend altering explanations to include a discussion that the compounds alter biomass instead of specifically biofilm viability.

2) The authors supplemented cells with exogenous AHL and stated the compound decreased biomass production. I don't see the defined connection between this compound and QS inhibition because the authors did not measure QS production in cells treated with the compound.

There are significant issues with readability due to errors in grammar/syntax. There are issues with the in-text citations as many of the citations come after the period but should be before. I saw a figure legend for figure 3, but did not see figure 3. Figures 2, 4, 7, and 9 are missing statistical analyses and/or p values.

Author Response

Due to recommendation of a referee the title of the article was changed from “Natural Inhibitors of Biofilm Formation and the Activity of the Autoinductor C6-AHL in Klebsiella pneumoniae ATCC 13884” to “The Search for Natural Inhibitors of Biofilm Formation and the Activity of the Autoinductor C6-AHL in Klebsiella pneumoniae ATCC 13884”. Similarly, there were modifications in the Introduction, Methodology, and Discussion. Finally, the original article was edited by a service of MDPI, however after that the authors added text and changed some content.

REFEREE 4

·         This approach has become a well-accepted model for biofilm formation:

o   Franklin et al. New Technologies for Studying Biofilms. Microbiol Spectr. 2015 August ; 3(4): . doi:10.1128/microbiolspec.MB-0016-2014.

o   Lebeaux. et al. From in vitro to in vivo Models of Bacterial Biofilm-Related Infections-Pathogens. 2013 Jun; 2(2): 288–356. doi: 10.3390/pathogens2020288

o   Goncalvez et al. Anti-Biofilm Activity: A Function of Klebsiella pneumoniae Capsular Polysaccharide. PLoS One. 2014; 9(6): e99995. doi: 10.1371/journal.pone.0099995

2. The authors supplemented cells with exogenous AHL and stated the compound decreased biomass production. I don't see the defined connection between this compound and QS inhibition because the authors did not measure QS production in cells treated with the compound.

·         Quorum Sensing (QS), is carried out through the production of autoinductor molecules, like C6-AHL; this substance is associated to the formation of biofilm. In this assay we want to establish whether 3-methyl-2 (5H)-furanone inhibits the effect of C6-AHL. For this, an experiment was done to determine the biofilm induced by pure C6-AHL; in another assay the effect of 3-methyl-2 (5H)-furanone pure was established, and a mixture of both substances was finally assayed.

Thus, Paragraph was written again:

2.11.Effect of 3-Methyl-2(5H)-Furanone on the Biofilm Inductor of C6-AHL.

In order to establish the effect 3-Methyl-2 (5H)-Furanone in the capacity of C6-AHL to induce the formation of biofilm, an assay was made in microplates of 96 wells with LB medium, according to the modified method of O´Toole [8]. Thus, C6-AHL at a final concentration of 0.4 µg/ml was placed. Similarly, in another wells 3-methyl-2(5H)-furanone at 15 µg/ml was added, and finally, in another, a mixture of both compounds at the described concentrations. Subsequently, an inoculum of K. pneumoniae adjusted to a OD600nm of 0.05 UA was added to the wells, and next it was incubated at 37 °C for 30 h. The measurement of the adhered bacterial mass was made by staining with violet crystal at 0.05% for 10 min, washed with distilled water and then dried for 2 h at 50 ° C, and extracted with 250 µ L of methanol dry again. The crystal violet absorbance (biofilm) was quantified at 585 nm in a microplate reader.

Results of this assays were reported in 3.8. Effects of 3-methyl-2(5H)-furanone against the C6-AHL Autoinductor.

3. There are significant issues with readability due to errors in grammar/syntax.

·         The original article was edited by a service of MDPI, however after that the authors added text and changed some content.

4. There are issues with the in-text citations as many of the citations come after the period but should be before.

·         It was corrected

5. I saw a figure legend for figure 3, but did not see figure 3.

·         It was corrected, since original figure was imbedded in the text

6. Figures 2, 4, 7, and 9 are missing statistical analyses and/or p values.

·         p values were included in figures

Round 2

Reviewer 2 Report

The article is improved, but its some editing needed in Figure 3: please fix y-axis title and numbering, it should start from 0 and not from negative numbers.

Author Response

An extensive review of the grammar and style have been made, some of which are highlighted in yellow color.

Concerning changes in figure 3, y-axis title was fixed, but it is not possible to modify numbering since several substances are biofilm inducer; thus, results are negative

Reviewer 4 Report

The manuscript would benefit from additional editing, but is improved over the previous version. The addition of statistical analyses is appreciated.

Author Response

An extensive review of the grammar and style have been made, some of which are highlighted in yellow color.